# Pickle Recruits Retinoblastoma Related 1 to Control Lateral Root Formation in *Arabidopsis*

**DOI:** 10.3390/ijms22083862

**Published:** 2021-04-08

**Authors:** Krisztina Ötvös, Pál Miskolczi, Peter Marhavý, Alfredo Cruz-Ramírez, Eva Benková, Stéphanie Robert, László Bakó

**Affiliations:** 1Department of Plant Physiology, Umeå Plant Science Center, Umeå University, S-901 87 Umeå, Sweden; 2Institute of Science and Technology Austria, Am Campus 1, 3400 Klosterneuburg, Austria; peter.marhavy@slu.se (P.M.); eva.benkova@ist.ac.at (E.B.); 3Bioresources Unit, AIT Austrian Institute of Technology, 3430 Tulln, Austria; 4Department of Forest Genetics and Plant Physiology, Umeå Plant Science Center, Swedish University of Agricultural Sciences, S-901 87 Umeå, Sweden; pal.miskolczi@slu.se (P.M.); Stephanie.Robert@slu.se (S.R.); 5Laboratory of Molecular and Developmental Complexity at Laboratorio Nacional de Genómica para la Biodiversidad, Centro de Investigación y de Estudios Avanzados del Instituto Politécnico Nacional, (CINVESTAV-IPN), 36590 Irapuato, Mexico; alfredo.cruz@cinvestav.mx

**Keywords:** chromatin remodeling, auxin signaling, *de novo* organogenesis

## Abstract

Lateral root (LR) formation is an example of a plant post-embryonic organogenesis event. LRs are issued from non-dividing cells entering consecutive steps of formative divisions, proliferation and elongation. The chromatin remodeling protein PICKLE (PKL) negatively regulates auxin-mediated LR formation through a mechanism that is not yet known. Here we show that PKL interacts with RETINOBLASTOMA-RELATED 1 (RBR1) to repress the *LATERAL ORGAN BOUNDARIES-DOMAIN 16* (*LBD16)* promoter activity. Since LBD16 function is required for the formative division of LR founder cells, repression mediated by the PKL–RBR1 complex negatively regulates formative division and LR formation. Inhibition of LR formation by PKL–RBR1 is counteracted by auxin, indicating that, in addition to auxin-mediated transcriptional responses, the fine-tuned process of LR formation is also controlled at the chromatin level in an auxin-signaling dependent manner.

## 1. Introduction

Lateral roots (LRs) are initiated in lateral root founder cells in the pericycle. In the model plant Arabidopsis (*Arabidopsis thaliana*) and many other species, LRs arise from pericycle cells opposite the xylem pole [1]. While in maize [2] and other grasses, including rice (*Oryza sativa*, [3]) and wheat (*Triticum vulgar*, [4]) LR founder cells are located at the phloem poles and in leptosporangiate ferns [5] at the xylem pole endodermis cells. The plant hormone auxin triggers the reprogramming of non-dividing cells into proliferative LR founder cells, then later acts in a gradient for the execution of further steps in the LR developmental program. This developmental process initiates by auxin signaling converging on protoxylem pericycle cells, which promotes the degradation of AUXIN/INDOLE-3-ACETIC ACID (Aux/IAA) proteins involved in LR initiation (LRI) [6]. Elimination of Aux/IAA repressors through SKP-Cullin-F-box (TIR1/AFB) (SCF^TIR1/AFB^) ubiquitin ligase complexes and the 26S proteasome results in activation of AUXIN RESPONSE FACTOR (ARF)7/ARF19 transcription factors to drive the expression of *LATERAL ORGAN BOUNDARIES DOMAIN/ASYMMETRIC LEAVES-LIKE (LBD/ASL) LBD16/ASL18*, *LBD29/ASL16* and many other target genes required for auxin response, LRI and development [7].

Shortly after auxin signal perception, a pair of pericycle cells adjacent to the xylem pole becomes polarized when nuclei of these cells migrate toward the common cell walls, thereby creating intracellular asymmetry [1]. Anticlinal division of such polarized cells yields two larger flanking and two smaller central daughter cells, the latter of which continue to divide periclinally to form the LR primordia [6]. Nuclear migration and establishment of asymmetry in LR founder cells is compromised in plants expressing a dominant negative version of LBD16, suggesting that LBD16 is one of the key players mediating formative cell division and LRI [8]. Polar nuclear movement and anticlinal cell division is inhibited in the gain-of-function *solitary-root* (*slr-1)* mutant expressing a non-degradable version of the SLR/IAA14 repressor protein; hence, the mutant lacks lateral roots [9]. Overexpression of CYCLIN D3;1, a known activating subunit of the G1/S regulator CDKA;1 kinase, triggers a few rounds of pericycle division but fails to initiate LR formation in the *slr-1* root [10]. Conversely, disruption of the *PICKLE* (*PKL*) gene encoding a chromodomain-helicase-DNA binding (CHD) ATP-dependent chromatin remodeling factor restores LR formation in the *slr-1* background indicating that inactivation of the *PKL* gene enables both the initial formative divisions as well as the subsequent organized proliferation of pericycle cells [11]. It has been therefore proposed that PKL negatively regulates LR initiation at the chromatin level; however, the mechanism through which PKL acts remained obscured.

PICKLE is a plant homologue of the animal chromatin remodeling ATPase Mi-2/CHD3/4 proteins, which in vertebrates form the Mi-2/nucleosome remodeling and deacetylase (NuRD) repressor complexes regulating chromatin organization, gene transcription and developmental signaling [12]. Animal NuRD complexes contain the ATPase chromatin remodeler CHD3/CHD4 proteins and a histone deacetylase subcomplex that comprises the histone deacetylase HDAC1/HDAC2 enzymes and the retinoblastoma-binding RbAp46 and RbAp48 histone chaperon proteins [13]. The presence of class 1-type histone deacetylases and a panel of RbAp46/48 homologues in the *Arabidopsis* genome suggests that, similar to animal systems, plant Mi-2/CHD3/4 ATPase remodelers might assemble to NuRD-like complexes. However, biochemical characterization of the *Arabidopsis* PKL protein failed to find evidence for the existence of such complexes thus far [14].

Intriguingly, the PKL protein sequence contains two LxCxE peptide motifs that are often present in viral and cellular proteins and mediate stable binding by fitting into a groove within the conserved small pocket domain of retinoblastoma (pRB) proteins. Animal retinoblastoma proteins and the plant ortholog RETINOBLASTOMA-RELATED 1 (RBR1) control the G1-to-S-phase progression in the cell cycle [15]. In the G1 phase, the hypophosphorylated form of pRB binds to and inactivates the E2F/DP1 transcription factor heterodimer, the activity of which is necessary for G1-to-S progression. Phosphorylation of pRB by cyclin-dependent kinase (CDK)/CyclinD complexes releases active E2F/DP1 dimers, initiates the transcription of S-phase specific genes and triggers cell division. Several lines of evidence indicate that the function of retinoblastoma proteins extends much beyond the canonical G1-to-S-phase control role. Human pRB protein has been implicated in cellular differentiation by associating with tissue-specific transcription factors and modulating their activity [16]. In vertebrates pRB is often present in chromatin repressor complexes that have roles in developmental transitions [17,18]. These findings strongly suggest that the pRB protein regulates cellular differentiation separate from its function in cell cycle progression [19]. Plant RBR proteins share the basic structural and functional features of pRB [15,20]. Similar to animal pRB, plant RBR proteins can associate with histone deacetylases to repress gene transcription [21]. While human pRB binds to histone deacetylases directly through the LxCxE [22], plant HDAC proteins do not contain the LxCxE motif, and accordingly, RBR proteins interact with HDACs indirectly. It has been reported that the *Arabidopsis* RBR1 binds to the MULTICOPY SUPPRESSOR OF IRA1 (MSI1) protein, which is a plant homologue of the animal RbAp46/48 proteins [23]. Evidence indicates that members of the plant MSI protein family associate with histone deacetylases to mediate transcriptional silencing at target loci [24]. The RBR1–MSI1 interaction takes place at the RbA pocket domain of RBR1, leaving the LxCxE binding cleft that is located on the RbB pocket domain available for protein binding [23]. This interaction topology enables RBR1 to recruit histone deacetylases and simultaneously associate with transcription factors and chromatin modifiers containing the LxCxE motif.

We report here that PKL interacts with the RBR1 protein in the *Arabidopsis* root. Consistent with this finding we show that similar to the PKL protein, RBR1 is a negative regulator of LR formation. Our data further demonstrate that PKL recruits RBR1 to the promoter of the LR-specific *LBD16* gene. When bound to the promoter the PKL–RBR1 complex acts as a transcriptional repressor of *LBD16* and negatively regulates LR formation. Through the IAA14/ARF7/ARF19 signaling pathway, auxin releases the PKL–RBR1 complex from the *LBD16* promoter, indicating that this novel, chromatin-level regulation of LR formation is tightly coupled to auxin signaling.

## 2. Results

### 2.1. PKL Interacts with Arabidopsis RBR1

To test whether PKL and RBR1 proteins interact, we expressed epitope-tagged versions of PKL and RBR1 proteins in *Arabidopsis* protoplasts and analyzed their interaction by co-immunoprecipitation (co-IP) assays. Our results demonstrate that RBR1 binds to the PKL protein (Figure 1A). To confirm interaction by an independent method, we performed bimolecular fluorescence complementation (BiFC) assays. Interaction between RBR1 and the full-length PKL were detected in the nucleus (Figure 1B). Finally, the PKL/RBR1 interaction between the endogenous PKL and RBR1 proteins was validated in planta by co-IP experiment in five-day-old wild-type seedling roots (Figure 1C). Since PKL harbors two LxCxE motifs, we tested epitope-tagged, truncated versions of PKL for RBR1 interaction to find similar affinity binding of the N- and C-terminal halves (Appendix A). We also examined whether mutation of the peptide motifs disrupted binding by converting both LxCxE motifs in the PKL sequence to AxAxA. BiFC interaction assays showed that mutations weakened but did not abolish interaction, indicating that in addition to the LxCxE motif other domains of PKL are also involved in contacting the RBR1 protein (Appendix A).

### 2.2. RBR1 Is Expressed in Xylem Pole Pericycle Cells

In *Arabidopsis* seedlings, PKL is expressed in meristems, organ primordia and in the stele, including the pericycle cells ([11,25], Appendix A), while RBR1 is abundant in proliferating tissues, including the shoot and root apical meristems, proximal part of young leaves and emerging LRs ([26], Appendix A). To explore whether the RBR1 protein is present in pericycle cells where PKL presumably functions to repress LR formation, we introduced the *pRBR:RBR-RED FLOURESCENT PROTEIN (RFP)* construct into the enhancer trap line J0192 in which GREEN FLOURESCENT PROTEIN (GFP) expression was restricted to xylem pole pericycle (XPP) cells [27]. Confocal microscopic analysis of roots expressing the RBR1–RFP fusion protein revealed that RBR1 was present in xylem pole pericycle cells (Figure 2). Interaction of PKL with RBR1 in the *Arabidopsis* root and the overlapping expression pattern of the two proteins in the differentiation zone indicated that RBR1 might participate in the PKL mediated repression of LR formation.

### 2.3. Similar to PKL, the RBR1 Protein Is a Negative Regulator LR Formation

Previous characterization of the *pkl/ssl2-1* mutant expressing a short N-terminal fragment of the PKL protein showed that the *ssl2-1* root is significantly shorter but produces LRs at similar density than the wild type [11]. However, under our experimental condition phenotypic analysis of *ssl2-1* confirmed the shorter root phenotype but indicated a significantly higher LR density compared with the wild type (Figure 3A,B and Appendix A). To assess if RBR1 had a role in LR development, we induced a reduction of RBR1 level in the diploid plant by partially silencing *RBR1* expression with the production of an artificial microRNA directed against the 3′ UTR of the *RBR1* mRNA [28]. Molecular analysis of the *amiRBR1* line showed that silencing decreased both transcript and protein levels by 50% compared with the wild type (Figure 3C,D). Phenotypic examination of the seedling roots revealed unperturbed primary root growth; however, an increased root branching was observed in the *amiRBR1* line (Figure 3E–G). To test whether root branching was sensitive to RBR1 abundance, we increased RBR1 level by expressing the RBR1-RED FLUORESCENT PROTEIN (*pRBR:RBR-RFP*) [29]. The presence of this construct in the *rbr2-1* background led to an increased expression of the wild-type gene, possibly due to the positive autoregulatory function of RBR1 over its own expression [30] (Figure 3C,D). Primary root growth was unaffected in the *RBR1-RFP* line, while root branching decreased compared with the wild-type seedling root (Figure 3E–G). Collectively, our data support a negative role for RBR1 in LR formation and indicate that an inverse relationship exists between RBR1 expression level and LR density. The fact that lower levels of both RBR1 and/or PKL enhanced LR formation and that the overlapped expression pattern of PKL and RBR1 (Appendix A) suggested a functional relationship between the two proteins in LR formation.

### 2.4. PKL Recruits RBR1 to the LBD16 Promoter

Polar nuclear movement and asymmetric divisions are blocked in LR founder cells of the *slr-1* mutant [31]. Because disruption of *PKL* restored LR formation in the *slr-1* root, the PKL protein probably acted to repress either polar nuclear movement or the subsequent asymmetric cell divisions. Consistent with a PKL–RBR1 functional interaction, the *RBR1-RFP* line produced fewer LRs but did not show aborted LR initiation events, indicating that RBR1 protein level also affected LRI at an early stage. Recent experimental data indicate that nuclear movement is mediated by the LBD16/ASL18 and related LBD/ASL transcription factors in the *Arabidopsis* root. Expression of a dominant negative repressor version of the LBD16 protein did not affect LR founder cell specification but prevented nuclear migration and abolished LR formation [8].

To examine whether the PKL and RBR1 proteins are involved in the transcriptional regulation of the *LBD* genes that are known to have a role in LR initiation, we performed chromatin immunoprecipitation (ChIP) assays. We designed primers specific for the promoter of *LBD16*, *LBD17*, *LBD18*, *LBD29* and *LBD33* genes and analyzed RBR1-bound chromatin samples by PCR. Out of the five *LBD* genes tested, ChIP with the RBR1 antibody pulled down DNA fragments of the *LBD16* promoter only (Figure 4A and Appendix A). Analysis of the *LBD16* promoter sequence revealed the presence of a consensus TTTGCCGG E2F binding motif 1374 bp upstream of the translation initiation site. Similar to pRB proteins of human and fly, *Arabidopsis* RBR1 binds to members of the E2F transcription factor family [32] and can be recruited to promoters containing the consensus E2F binding site sequence [33]. We therefore tested if the interaction of RBR1 with the *LBD16* promoter was mediated by E2F transcription factor. Our data showed this was not the case, fragment containing the predicted E2F binding site could not be detected in ChIP samples. By contrast, the promoter proximal fragments could be amplified, indicating that binding took place close to the transcription start site (TSS) (Fragments F2 and F4 in Figure 4A).

Interaction between PKL and RBR1 proteins suggested that they act on a common pathway to regulate LR formation; thus, the presence of PKL on the *LBD16* promoter was tested by ChIP assay. Quantitative real-time PCR (qRT-PCR) analysis of PKL-bound chromatin samples revealed that PKL bound to the *LBD16* promoter and that binding took place within the same proximal promoter region (Fragment F4) to which RBR1 binds (Figure 4B). To assess whether RBR1 binds directly or through interaction with PKL to the *LBD16* promoter, we examined promoter binding in a *ssl2-1* root that lacked the full-length PKL protein (Appendix A). ChIP with the anti-RBR1 antibody failed to show any enrichment of the *LBD16* promoter fragment (Figure 4C). That PKL interacted with RBR1 and promoter targeting was abolished when the PKL protein was absent collectively indicate that binding of the complex to the *LBD16* promoter took place through PKL.

### 2.5. Transcriptional and Functional Analysis of the Effect of RBR1 and PKL Proteins on LBD16 Expression

To assess the effect of PKL and RBR1 proteins on the *LBD16* promoter activity, we quantified *LBD16* gene expression in wild-type and mutant roots. *LBD16* expression was increased in the *amiRBR1* and *ssl2-1* lines and attenuated in the *RBR1-RFP* line, suggesting that RBR1 and PKL proteins act as repressors of the *LBD16* promoter activity (Figure 4D). The observed negative correlation between the abundance of PKL and RBR1 proteins and *LBD16* gene expression led us to test whether a similar relationship existed between protein levels and asymmetric cell division. Asymmetric divisions in roots of *amiRBR1*, *RBR1-RFP* and *ssl2-1* lines were quantified by analyzing the expression of the *ARABIDOPSIS CRINKLY4* (*ACR4)* gene that is a marker of formative pericycle divisions during LRI [34]. *ACR4* transcript levels were increased in the *amiRBR1* and *ssl2-1* lines, while they were decreased in the *RBR1-RFP* line (Figure 4D), indicating that by binding to the *LBD16* promoter PKL and RBR1 proteins ultimately regulated asymmetric cell division at LRI.

To exclude the possibility that altered RBR1 and PKL abundance would result in misexpression of *LBD16,* we introduced the *pLBD16.:GFP* construct into the *amiRBR1*, *RBR1-RFP* and *ssl2-1* lines. The *LBD16* expression domain was unaffected (Appendix A); however, the basal activity of the *LBD16* promoter in the *amiRBR1* and *ssl2-1* lines was strongly increased along the stele (Figure 5A). Consistent with the qRT-PCR (Figure 4D) and the LR density data (Figure 1B and Figure 3G), elevated *LBD16* promoter activity gave rise to increased LR density (Figure 5A, asterisks). Higher basal activity of the *LBD16* promoter in the *amiRBR1* and *ssl2-1* lines might have been due to an enhanced auxin activation response brought about by a less repressed state of the promoter. To test this hypothesis, we performed root bending assays to change the auxin distribution and concentration, which in turn induces LR formation as a consequence of the gravitropic stimulus [1,35]. Expression of the GFP reporter was much stronger in *amiRBR1* and *ssl2-1* background compared with wild type, indicating that activation of the *LBD16* promoter was enhanced when either RBR1 or PKL protein levels were low and the promoter was not subjected to repression at the chromatin level (Figure 5B,C).

### 2.6. Chromatin Context Is Required for Proper Control of LBD16 Promoter Activity

To study PKL–RBR1 repressor binding to the *LBD16* promoter in a system where auxin levels could be easily manipulated, we expressed a *pLBD16::GFP* reporter construct in suspension culture-derived protoplasts. In the absence of auxin, a strong basal activity of the *LBD16* promoter was detected, and this basal activity was not induced when transfected cells were cultured in the presence of auxin (Figure 6A). By contrast, the well-established auxin-signaling reporter *pDR5rev::GFP* [36] showed the expected auxin inducible expression pattern in transfected protoplasts (Figure 6A). Activity of the *LBD29* promoter, another LR specific gene, as well as expression of the endogenous *LBD16* gene were also inducible by auxin, indicating that the auxin signaling pathway was functional in transfected protoplasts (Figure 6A,D). Deletion of the distal part of the *LBD16* promoter (from −1547 to −811 bp) containing binding sites for E2F and ARF transcription factors decreased its activity by about 60% while deletion of the proximal part of the promoter (from −811 to +1 bp) reduced its activity by 90% relative to the full-length promoter (Figure 6B). Our prior data indicated that the PKL–RBR1 complex binds to the *LBD16* promoter and represses *LBD16* expression to control LR formation. Therefore, the high basal activity of the *pLBD16::GFP* transgene in transfected protoplasts could be due to the failure of the PKL–RBR1 repressor complex to bind the plasmid resident *LBD16* promoter. To test this hypothesis, we conducted transient plasmid ChIP assays [37] and analyzed the recovered DNA samples by using primers specific either to the plasmid-born or to the genomic *LBD16* promoter. Compared with the genomic promoter, almost no RBR1 binding was detectable on the extrachromosomal promoter (Figure 6C), indicating that the PKL–RBR1 repressor complex did not bind to the plasmid-born promoter. Overall, these data demonstrate that outside of its chromatin context (when residing on a plasmid that lacks a histone core) *LBD16* promoter activation was independent of an auxin-mediated signaling pathway and suggest that the activity and auxin responsiveness of the *LBD16* promoter was regulated at the chromatin level in its genomic context.

### 2.7. Auxin Signaling Is Required to Dissociate the PKL–RBR1 Complex from the LBD16 Promoter

Our experimental data indicate that PKL and RBR1 proteins interacted in the *Arabidopsis* root and that the complex associated with the *LBD16* promoter to repress *LBD16* gene expression and asymmetric cell division in the pericycle. According to the current model, *LBD16* promoter activity was regulated by auxin signaling through the SCF^TIR1^/IAA14/ARF7/19 pathway. How the PKL–RBR1 mediated repression mechanism integrated into this canonical model of transcriptional control and how auxin signaling was linked with chromatin level regulation of *LBD16* gene expression was unclear. To address these questions, we first tested whether TIR1/AFB receptor function was required by blocking the formation of the TIR1–IAA–Aux/IAA complexes with the auxin antagonist auxinol [38]. Treatment of seedlings with 20 µM auxinol entirely prevented LRI in all lines, indicating that even if limiting PKL or RBR1 protein levels precluded repressor complex formation, activation of the *LBD16* promoter and LRI still required auxin action (Appendix A).

Next, we investigated how the PKL–RBR1 mediated repression of the *LBD16* promoter responded to auxin signaling. We took advantage of the LR-inducible system in which application of exogenous auxin activates the LR initiation program in all pericycle cells [39] and consequently, induced *LBD16* expression in a synchronized manner (Figure 7A). Activation of LRI enabled us to examine changes of *LBD16* promoter occupancy by using molecular tools. Treatment of 10 days after germination (DAG) wild-type seedling roots with 10 µM of the synthetic auxin 1-Naphtaleneacetic acid (NAA) for 2 h reduced binding of RBR1 and PKL proteins to the *LBD16* promoter by 50% (Figure 7B). Conversely, we could only detect subtle changes in the dominant negative auxin signaling mutant *slr-1* root upon auxin treatment. Because of the lower expression level of RBR1 and PKL in the *slr-1* root (Appendix A), binding to the *LBD16* promoter appeared generally weaker. Our data indicate that auxin signaling dissociated the RBR1-PKL complex from the *LBD16* promoter and that *LBD16* expression depended on the coordinated action of chromatin remodeling factors and the auxin signaling pathway.

## 3. Discussion

Previous studies have identified the essential signaling response modules through which the phytohormone auxin initiates LR formation. The first module consists of the SLR/IAA14-ARF7-ARF19 proteins and reactivates the cell cycle as well as controls the first asymmetric divisions of the pericycle. The BODENLOS (BDL)/IAA12- MONOPTEROS (MP)/ARF5 pair forms the second auxin response module that shares some of the functions of the first module but also acts to prevent further unsolicited formative divisions [1,40,41,42]. In both modules, binding of auxin to the TIR1/AFB receptors results in degradation of the Aux/IAA repressors of cognate ARF transcription factors and activates gene transcription. We show here that parallel to the destabilization of signaling repressors auxin also acts to disengage a repressor mechanism that controls *LBD16* promoter activity at the chromatin level. This repressor mechanism contains the ATP-dependent chromatin remodeler PKL that associates with the RBR1 protein in the root. The PKL–RBR1 complex binds to and represses the activity of the LR-specific *LBD16* gene promoter in an auxin-dependent manner and in turn restricts the first asymmetric cell divisions of the pericycle. Our data establish a link between auxin signaling and chromatin level regulation and reveal a novel mechanism underlying LRI in *Arabidopsis*.

### 3.1. PKL Recruits RBR1 to the LBD16 Promoter to Form a Repressive PKL–RBR1 Complex

As an ATP-dependent chromatin remodeler of the CHD family, the PKL protein sequence harbors PHD (PLANT HOMEODOMAIN) finger and chromo domains as well as SANT-SLIDE domains (Appendix A) [14], each of which has been implicated in chromatin binding [43]. The N-terminus of the PKL protein contains two tandem chromo domains that have been proposed to bind methylated lysines of the histone H3 tail, including H3K27me3 [44]. In animals and in plants, the Polycomb repressive complex 2 (PRC2) mediates H3K27 trimethylation, which is an important repressive mark with critical roles in developmental processes [45]. In *pkl* plants, deposition of the H3K27me3 mark is significantly reduced, suggesting that PKL itself somehow promotes H3K27me3 modification at target loci [46]. Recent ChIP data also show that PKL is present at these target loci; however, many of these genes show PKL independent expression [47]. Genome-wide mapping of histone H3K27me3 marks in *Arabidopsis* indicates that *LBD16*, *LBD17*, *LBD18*, *LBD29* and *LBD33* loci are all decorated by H3K27me3 [48,49]. That in ChIP assays we detected binding of the PKL–RBR1 complex exclusively to the *LBD16* promoter suggests that targeting involves a mechanism other than the histone code reader function of the PKL chromo domain. PKL has also been identified as a negative regulator of photomorphogenesis by binding to the transcription factors ELONGATED HYPOCOTYL5 (HY5)/HY5-HOMOLOG (HYH) [50]. Targeting of PKL to promoters of cell elongation-related genes is compromised in the *hy5hyh* double knock-out mutant, indicating that, notwithstanding its DNA binding domains, recruitment of PKL to definite gene loci requires association with sequence-specific transcription factors. We thus hypothesize that the observed selectivity of the RBR1-PKL complex for the *LBD16* promoter might also be mediated by a PKL-associated transcription factor. Our ChIP assays showed enrichment of F2 and F4 genomic fragments that are encompassing the −790 to +36 bp region from the proximal part of the promoter, indicating that binding of the complex takes place near the TSS. Positioning of nucleosomes around the 5′ end of many eukaryotic genes shares common organizational features, according to which the TSS is located in a nucleosome-depleted region flanked by two arrays of nucleosomes [51]. The pattern of nucleosome spacing and distribution is established and maintained by ATP-dependent chromatin remodeler complexes. Activity of the CHD1 chromatin remodelers Hrp1 and Hrp3 is required in the fission yeast to maintain nucleosome organization in gene coding regions [52], whereas in the budding yeast, the CHD3-CHD4 remodeler Mit1 plays a similar role [53]. In *Arabidopsis,* the SWI/SNF chromatin remodeling ATPase BRAHMA (BRM) regulates abscisic acid responses. In the absence of ABA signaling, BRM keeps the ABA-responsive *ABI3* and *ABI5* genes in a repressed state by binding to their promoter close to the TSS. Accordingly, BRM has been proposed as facilitating high occupancy of the +1 nucleosome adjacent to the TSS at the *ABI5* locus [54]. The TSS proximal occupancy of the PKL–RBR1 complex indicates that PKL might act in a similar manner to establish a repressive nucleosomal landscape at the *LBD16* locus. Positioning of the PKL–RBR1 complex is also consistent with recent genome-wide analyses of Rbf1 binding sites in *Drosophila* and RBR1 in Arabidopsis, showing strong promoter-proximal targeting bias for both proteins [55,56].

### 3.2. Auxin Signaling Regulates LBD16 Expression by Two Distinct but Interconnected Mechanisms

Because LBD16 function mediates the first asymmetric divisions of pericycle cells, the mechanism controlling *LBD16* gene expression is of great importance for LR formation. Prior data have shown that auxin signaling initiates degradation of IAA14 repressor protein and in turn promotes dimerization of ARF7–ARF19 transcription factors that ultimately drive *LBD16* expression by binding to AuxRE cis- elements on the promoter [57]. This mechanism of promoter regulation can work as an on–off switch, provided that the region between enhancer and TSS—and in particular the TSS-adjacent core promoter—is inherently silent in the absence of auxin. Our data however show that the proximal part of the *LBD16* promoter possesses fairly high basal activity, indicating that the core promoter encompassing the TSS can initiate transcription without the trans-acting function of ARFs. Therefore, an additional auxin responsive molecular mechanism must be present to keep tight control over the basal activity of the core promoter. We propose that the PKL–RBR1 complex fulfills this role. Both PKL and RBR1 proteins bind to the same proximal part of the promoter, the two proteins associate in vivo and the complex represses *LBD16* transcription and LR formation.

We further show that chromatin level regulation of the *LBD16* promoter activity acts in concert with auxin signaling because reduced PKL or RBR1 level alone is not sufficient for LR initiation and derepression of the *LBD16* promoter requires auxin and a functional SLR/ARF7/19 auxin signaling mechanism. Our data support that, parallel to the SLR/ARF7/19 transcriptional module, expression of the *LBD16* gene is controlled by the chromatin-bound PKL–RBR1 repressor complex. This dual regulation ensures that critical asymmetric cell divisions occur only upon auxin signal perception that is strictly coordinated with chromatin remodeling activities. The PKL–RBR1 complex represses *LBD16* promoter activity in an auxin-dependent fashion and in turn regulates asymmetric cell division. PKL binds either directly or through a sequence-specific transcription factor to the *LBD16* promoter. In the absence of auxin signaling, PKL binds RBR1 that can recruit a chromatin modifier to the complex. Auxin signaling dissociates the complex probably by inducing post-translational modifications of either PKL or RBR1 proteins. RBR1 is hyperphosphorylated at the onset of LR formation (our unpublished data), and recent phosphoproteomic analysis of auxin-induced LR formation in *Arabidopsis* revealed that PKL is a phosphoprotein, and its phosphorylation status changes upon application of the phytohormone [58]. It has been reported that the *Drosophila* dMi-2/CHD3 protein is phosphorylated by casein kinase 2 (dCK2) at the N-terminus and that phosphorylation modulates its nucleosome binding and ATP-dependent nucleosome mobilization activities [59]. Remarkably, the PKL N-terminal sequence contains a consensus CK2 phosphorylation site within the PHD domain, and phosphoproteomic data indicate this site is phosphorylated in the root [58]. It is thus possible that, similar to dMi-2, PKL activity and chromatin association is also under post-translational control. We observed that a fraction of PKL remains bound to the promoter, perhaps to prevent deposition of H3K27me3 marks, or alternatively, due to its ATP-dependent remodeling activity, PKL alters the position of nucleosomes. Our findings presented here give the first insight into how chromatin level *regulation* of a key developmental gene is integrated with auxin signaling to control formative cell divisions and lateral organ formation

## 4. Materials and Methods

### 4.1. Plant Material, Growth Conditions and Generation of Transgenic Lines

Seeds of *Arabidopsis thaliana* Columbia-0 wild type, *amiRBR1* [60], *RBR1-RFP* [29], *ssl2-1* [11], *slr-1* [9], *pPKL::PKL-GFP* in *pklpkr2* background [61] and pLBD16::GFP in Columbia-0, *amiRBR1*, *RBR1-RFP* and *ssl2-1* background were used. All seeds were grown in soil at 22 °C 16 h light (150 µmol m^−2^ s^−1^), and 18 °C (8 h dark) at 60% relative humidity. For aseptic growth, seeds were sterilized for 1 min with 70% (v/v) ethanol, soaked in NaOCl solution (1.2% NaOCl, 0.05% Triton X-100) for 7 min, washed three times with sterile deionized water and plated on 1% agar plates containing 0.5× Murashige and Skoog salt mixture including vitamins (Duchefa) and 0.5% Sucrose. The plates were maintained in darkness at 4 °C for 2 days for stratification and then placed for 10 days at 22 °C 16 h light (150 µmol m^−2^ s^−1^) and 18 °C (8 h dark) at 60% relative humidity. Five days after germination (DAG), roots in vast quantity were harvested by cutting the roots of 5 DAG seedlings produced by using a hydroponic culture system, kept at 22 °C 16 h light (150 µmol m^−2^ s^−1^) and 18 °C (8 h dark) at 60% relative humidity. Transgenic *Arabidopsis* plants were generated by *Agrobacterium*-mediated transformation using the floral dip method.

### 4.2. Generation of Plasmid Constructs

All plasmid constructs were made using standard cloning techniques and polymerase chain reaction (PCR). Original cDNA clones were purchased from the Riken cDNA collection (http://www.brc.riken.jp, accessed on 25 March 2012). The N-terminal part of the PKL protein (aa 1-586) was synthetized by GenScript (http://www.genscript.com, accessed on 25 March 2013). Details of the molecular cloning work are provided in the Appendix A.

### 4.3. Production of Antibodies

Anti-RBR1 antibody was generated using a C-terminal polypeptide fragment of the RBR1 protein that was expressed in *Escherichia coli* as described earlier [62]. The purified recombinant protein was used to immunize rabbits by a company (Eurogentec). From the immunserum, a crude IgG fraction was isolated by ammonium sulfate precipitation, then IgG was further purified on protein gel blots of the antigen. To produce the PKL antibody, a C-terminal polypeptide encompassing the last 284 amino acids of the PKL protein was expressed in fusion with a hexahistidine tag in the *E. coli* strain BL21DE3 Rosetta (Novagen). The protein was purified under denaturing conditions on Ni^2+^-nitrilotriacetic acid matrix (Ni-NTA Agarose, Qiagen) and used to raise polyclonal antibody in rabbits (Eurogentec). From the crude immunserum, specific IgG fraction was isolated as described above.

### 4.4. Microscopy

For BiFC assays, Yellow Fluorescent Proteins (YFP) were visualized using a Leica TCS-SP2 laser scanning confocal microscope (LSM) (Leica Microsystems, Heidelberg, Germany) with a 63 × 1.4 NA oil-objective lens and processed using Leica confocal software v.2.61 about 16 h post-transformation. YFP was excited with a 514 nm argon laser, and fluorescence was detected between 520–550 nm. For *LBD16* promoter activity assays, fluorescence detection of Green Fluorescent Proteins (GFP) was performed using an LSM 510, AxioObserver (Carl Zeiss, Germany) microscope with a 10 × 0.45 M27 objective lens. GFP was excited at 488 nm, and the emitted light was captured at 505 to 555 nm. GFP fluorescence intensities captured by ZEN software (Carl Zeiss, Germany) were measured and quantified by ImageJ software (http://imagej.nih.gov/ij/, accessed on 25 March 2013).

### 4.5. Transcript Level Analysis

Roots of 10-day-old seedlings or suspension derived protoplasts were harvested, and total RNAs were extracted using the RNeasy kit (Qiagen). For quantitative RT-PCR, RNAs were treated with DNaseI (Ambion) and reverse transcribed using the first-strand cDNA synthesis kit (iScript Advanced cDNA Kit, BioRad). Gene-specific primers and iQSYBER Green Supermix (BioRad) were used on a C1000 Thermal Cycler (BioRad). Quantitative RT-PCR was performed using three replicates, and At1g61670, At3g03210 and At2g32760 were chosen as reference genes [63,64]. Amplification cycles were analyzed using the Bio-Rad CFX Manager Software (Version 1.6), and fold expression change for each gene of interest was calculated using the ΔΔCt comparative method. Primer sequences are listed in Appendix A.

### 4.6. Transient Expression of Proteins in Suspension Derived Protoplasts and Immunoprecipitation

For all experiments, an *Arabidopsis* cell suspension derived from wild-type *Col-0* roots [65] was used. Protoplast isolation and transient expression experiments were done according to [66], with slight modifications: 5 × 10^5^ protoplasts were used for each transformation with 3 to 5 μg plasmid DNA. After transfection, protoplasts were incubated in the dark for 16–24 h, harvested by centrifugation, then proteins were extracted from the cell pellet as described previously [60].

For the coimmunoprecipitation assays, 100 μg total protein extract was incubated in a total volume of 100 μL extraction buffer containing 150 mM NaCl and 1 μg anti-HA or 1.5 μg anti-cMyc or 2 μg anti-RBR1 antibody (Covance, clone 16B12 for HA and 9E10 for c-Myc, respectively). After 2 h, 10 μL Protein G-Sepharose matrix (GE Healthcare) was added, which was previously equilibrated in TBS buffer, and this mixture was further incubated for another 2 h on a rotating wheel at 4 °C. The matrix was then washed in 3 × 500 μL washing buffer (1 × TBS, 5% glycerol, 0.1% Igepal CA-630) and eluted by boiling in 25 μL 1.5 × Laemmli sample buffer. Proteins were then resolved with SDS-PAGE and blotted to PVDF transfer membrane (Millipore). The presence of the proteins of interest was tested by immunodetection using rat anti-HA-peroxidase (3F10, Roche) or chicken anti-c-Myc primary antibody (A2128, Invitrogen), rabbit anti-chicken IgY HRP conjugate (Thermo Scientific) and anti-PKL antibody, respectively.

### 4.7. Chromatin Immunoprecipiation (ChIP) Assays

ChIP analysis was performed as described [67,68]. Roots of ten- and five-day-old *Arabidopsis thaliana* Columbia-0 and roots of ten-day-old *slr-1* and *ssl2-1* seedlings were harvested, and proteins were cross-linked to DNA with 1% formaldehyde for 15 min. ChIP assays were performed using antibodies against PKL, RBR1 and normal rabbit IgG (Santa Cruz Biotechnology). Gene-specific primers and *SsoAdvanced*
*SYBR*
*Green* Supermix (BioRad) were used on a C1000 Thermal Cycler (BioRad). Quantitative ChIP-PCR was performed in three replicates, and results were analyzed according to the percentage input method or to the fold enrichment method [69]. ChIP experiments were performed at least twice. Primer sequences are provided in Appendix A.

For plasmid ChIP, analysis protoplasts were prepared as described above, and ChIP assays were performed as described in [37,70].

## Figures and Tables

**Figure 1 ijms-22-03862-f001:**
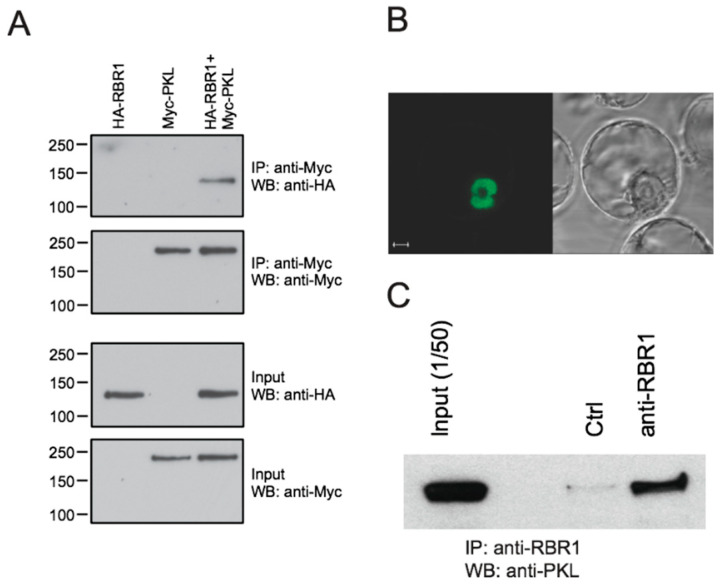
Chromatin remodeling protein PICKLE (PKL) interacts with RETINOBLASTOMA-RELATED 1 (RBR1) in vivo. (**A**) Protoplasts were transfected with *35S::HA-RBR1* (HA-RBR1) and *35S::myc-PKL* (Myc-PKL) constructs, and protein extracts from transformed protoplasts were immunoprecipitated with anti-myc antibodies. Immunocomplexes and input proteins were analyzed on protein gel blots using anti-HA or anti-myc antibodies, respectively. (**B**) Confocal microscopic image of the subcellular localization of the RBR1/PKL complex by bimolecular fluorescence complementation (BiFC) assays. Coexpression of *35S::YFPN-RBR1* (RBR1) and *35S::YFPC-fPKL* (fPKL) in suspension derived *Arabidopsis* protoplasts. Bar = 5 μm. (**C**) Co-immunoprecipitation (CoIP) assay in 5 days after germination (DAG) wild-type Columbia (Col-0) roots. Protein extracts were immunoprecipitated either with preimmune serum (Ctrl) or with anti-RBR1 antibody, and immunocomplexes were analyzed on protein gel blots using anti-PKL antibody. Input is 1/50 of the total protein amount used for the immunoprecipitation reactions.

**Figure 2 ijms-22-03862-f002:**
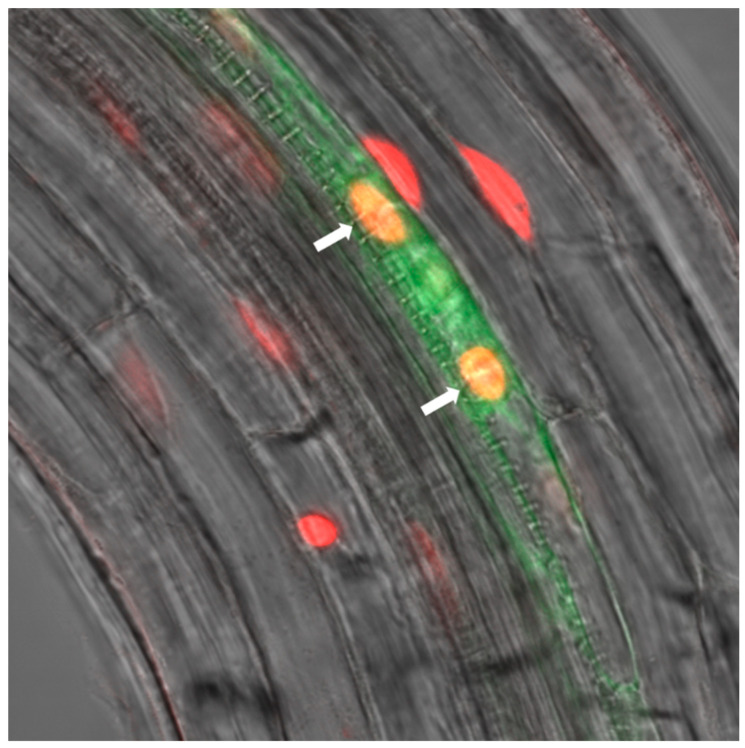
The RBR1 protein is present in lateral root (LR) founder cells in the pericycle. Laser scanning microscope image of J0192::*pRBR:RBR-RED FLOURESCENT PROTEIN (RFP)* root showing RBR-RFP fluorescence (red) and *pLBD16::GREEN FLOURESCENT PROTEIN (GFP)* fluorescence (green) marking lateral root founder cells in the pericycle. White arrows mark the position of the LR founder cells.

**Figure 3 ijms-22-03862-f003:**
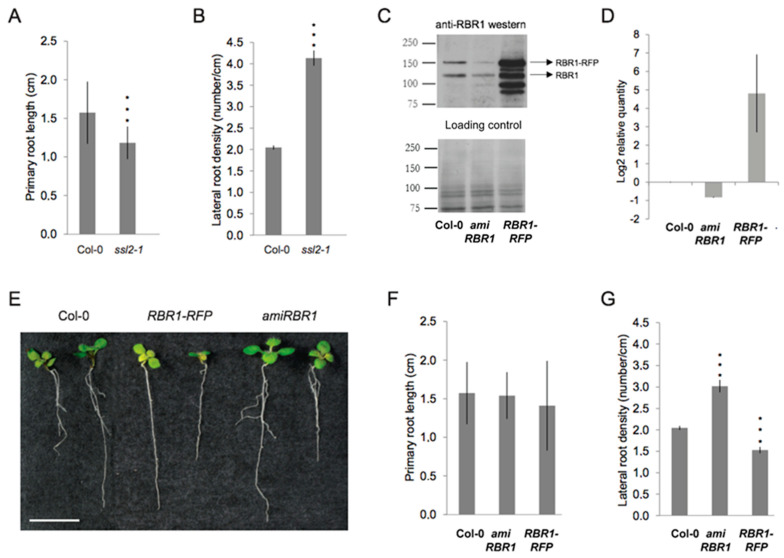
RBR1 protein abundance affects lateral root development. (**A**,**B**) Phenotypic analysis of 10 DAG *ssl2-1* roots. Quantification of the primary root length (**A**) and the lateral root density (**B**) in 10 DAG Col-0 and *ssl2-1* line. Error bars represent means ± SD of the mean. The data were normalized to the levels in Col-0, *p* < 0.001 by two-sided *t*-test; *n*_Col-0_ = 57, *n_ssl2-1_* = 41. (**C**) Western blot analysis of RBR1 protein level and (**D**) quantitative real time RT-PCR (qRT-PCR) analysis of *RBR1* mRNA expression in wild-type Columbia (Col-0), *amiRBR1* and *pRBR1::RBR1-RFP* (*RBR1-RFP*) lines. The expression level of *RBR1* was normalized to At1g61670 and At3g03210 (mean stability value *M* = 0.0405), and data (log2) from three technical replicates are shown as means ± standard deviation (SD). (**E**) Phenotype of 10 DAG seedlings expressing different levels of RBR1. Bar = 1 cm. (**F**) Quantification of the primary root length (cm) and (**G**) the lateral root density in 10 DAG Col-0, *amiRBR1* and *RBR1-RFP* roots. Error bars represent means ± SD of the mean. The data were normalized to the levels in Col-0, *p* < 0.001 by two-sided *t*-test (***); *n*_Col-0_ = 57, *n_amiRBR1_* = 50, *n_RBR1-RFP_* = 52.

**Figure 4 ijms-22-03862-f004:**
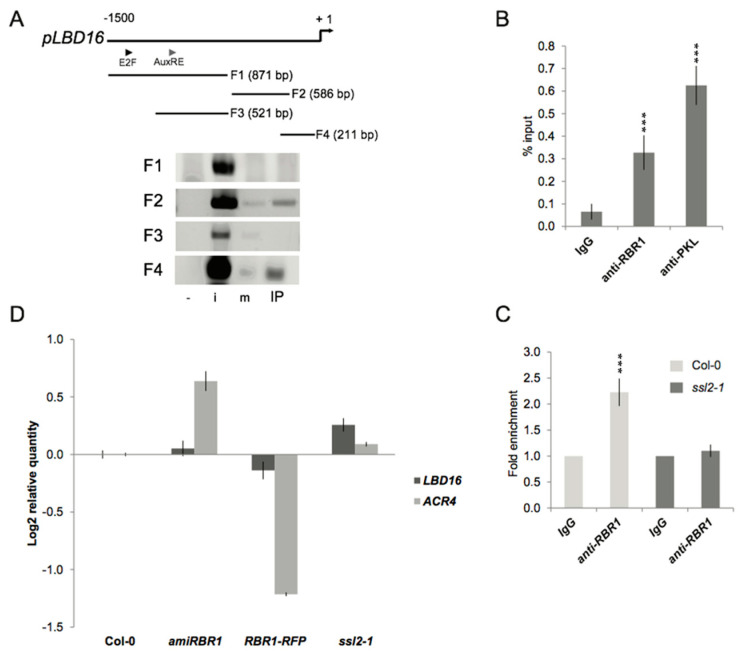
PKL recruits RBR1 to the *LBD16* promoter. (**A**) Chromatin immunoprecipitation (ChIP)-PCR analysis of different *LBD16* promoter fragments (F1–F4) precipitated with RBR1 antibody using chromatin extracted from 10 DAG Col-0 roots. Triangles show the position of the predicted E2F consensus (E2F) binding site, the AuxRE motif and the transcription start site (TSS); +1 labels the translation initiation site. -: non-template control, i: input DNA, m: IP with IgG, IP: IP with anti-RBR1 antibody. (**B**) ChIP-qRT-PCR analysis of RBR1 and PKL binding to *LBD16* promoter fragment (F4) in 5 DAG wild-type Columbia roots. Bar graphs show quantification of qRT-PCR products from ChIP experiment using anti-RBR1 and anti-PKL antibody, expressed as % input. Data from two biological replicates are shown as means ± SD of the mean; significance was evaluated by two-sided *t*-test at *p* < 0.001 (***) (**C**) ChIP-qRT-PCR analysis of RBR1 binding to the *LBD16* promoter in *ssl2-1* background. Bar graphs show quantification of qRT-PCR products from ChIP experiment using anti-RBR1 antibody on chromatin extracted from 10 DAG Col-0 and *ssl2-1* roots. Data from two biological replicates are shown as means + SD of the mean; significance was evaluated by two-sided *t*-test at *p* < 0.001 (***). (**D**) qRT-PCR analysis of *LBD16* and *ACR4* gene expression in 10 DAG roots expressing *35S::amiRBR1* (*amiRBR1*), *pRBR1::RBR1-RFP* (*RBR1-RFP*) and in the *PKL* mutant *ssl2-1* compared with Col-0. The expression levels were normalized to At1g61670 and At3g03210 (mean stability value *M* = 0.0413), and data (log2) from three technical replicates are shown as means ± SD.

**Figure 5 ijms-22-03862-f005:**
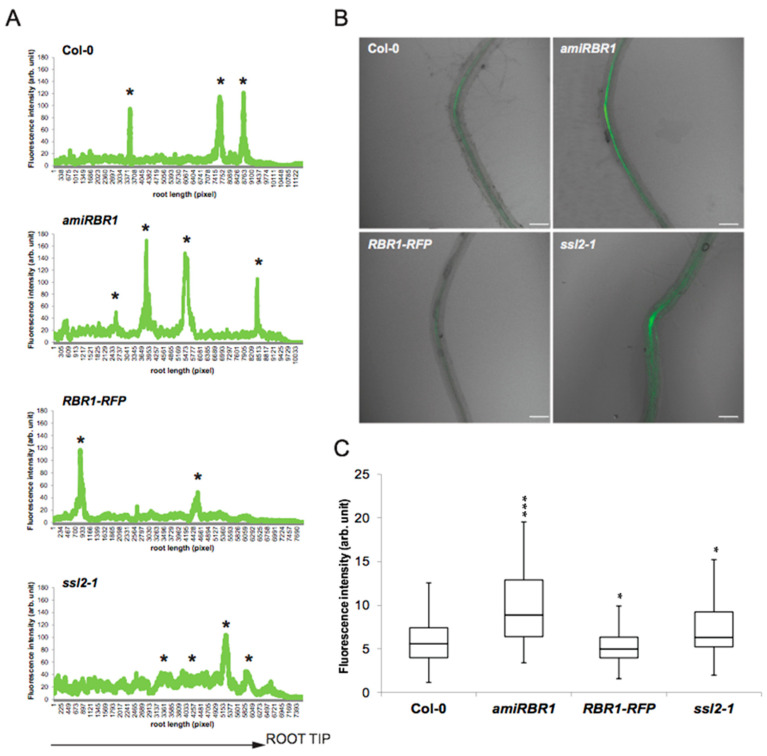
Functional analysis of the effect of RBR1 and PKL proteins on *LBD16* promoter. (**A**) Activity of the *pLBD16:GFP* reporter construct in 6 DAG roots of Col-0, *amiRBR1*, *RBR1-RFP* and *ssl2-1* lines. The graph shows GFP fluorescence intensity (arbitrary units) along the stele versus root length (in pixels). Asterisk (*) represents lateral root primordia/lateral roots. (**B**) Result of the root bending test. The confocal images show representative pictures from the bended sites of roots of different lines expressing *pLBD16::GFP* after 5 h following the 12 h long gravistimuli, and (**C**) the quantified data are summarized in box plots. The intensity of GFP fluorescence (arbitrary units) was measured in at least 31 roots per line; significance was evaluated by two-sided *t*-test at *p* < 0.05 (*) or *p* < 0.001 (***). *n*_Col-0_ = 39, *n_amiRBR1_* = 41, *n_RBR1-RFP_* = 31, *n_ssl2-1_* = 35. The scale bar on (**B**) = 100 μm.

**Figure 6 ijms-22-03862-f006:**
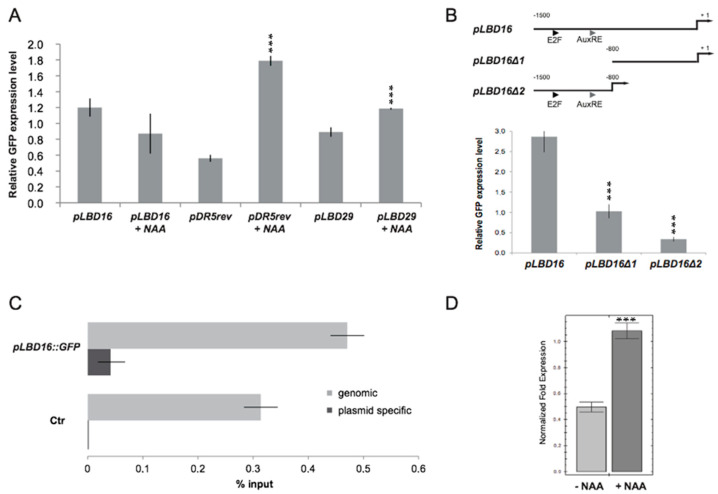
The *LBD16* promoter-reporter construct shows high and auxin-independent activity in *Arabidopsis* protoplasts. (**A**) GFP protein expression levels in suspension derived protoplasts treated with or without 10 μM NAA expressing the auxin responsive reporter *pDR5rev::GFP*, the lateral root specific *pLBD16::GFP* and *pLBD29::GFP* reporter constructs. Data from at least three biological replicates are shown as means ± SD of the mean; significance was evaluated by two-sided *t*-test at *p* < 0.001 (***). (**B**) *LBD16* promoter deletion assay. GFP protein expression levels in suspension derived protoplasts expressing the *pLBD16::GFP* (*pLBD16*), the *pLBD16Δ1::GFP* (*pLBD16Δ1*) and *pLBD16Δ2::GFP* (*pLBD16Δ2*) constructs. The deleted regions are shown in the diagram. Triangles mark the position of the predicted E2F consensus (E2F) and the AuxRE motives; +1 labels the translation initiation site. Significance was evaluated by two-sided *t*-test at *p* < 0.001 (***). (**C**) ChIP-qRT-PCR analysis of RBR1 binding to genomic and plasmid-born *LBD16* promoter using chromatin isolated from protoplasts transfected either with control (Ctr) or with *pLBD16::GFP* construct. Bar graphs show quantification of qRT-PCR products from ChIP experiment using anti-RBR1 antibody, expressed as % input. Data from two biological replicates are shown as means ± SD of the mean. (**D**) qRT-PCR analyses of the endogenous *LBD16* gene expression level in the presence and in the absence of 10 μM NAA. The expression level of *LBD16* was normalized to At2g32760 and At3g03210 (mean stability value *M* = 0.0398), and data (linear) from three technical replicates are shown as means ± SD; significance was evaluated by two-sided *t*-test at *p* < 0.001 (***).

**Figure 7 ijms-22-03862-f007:**
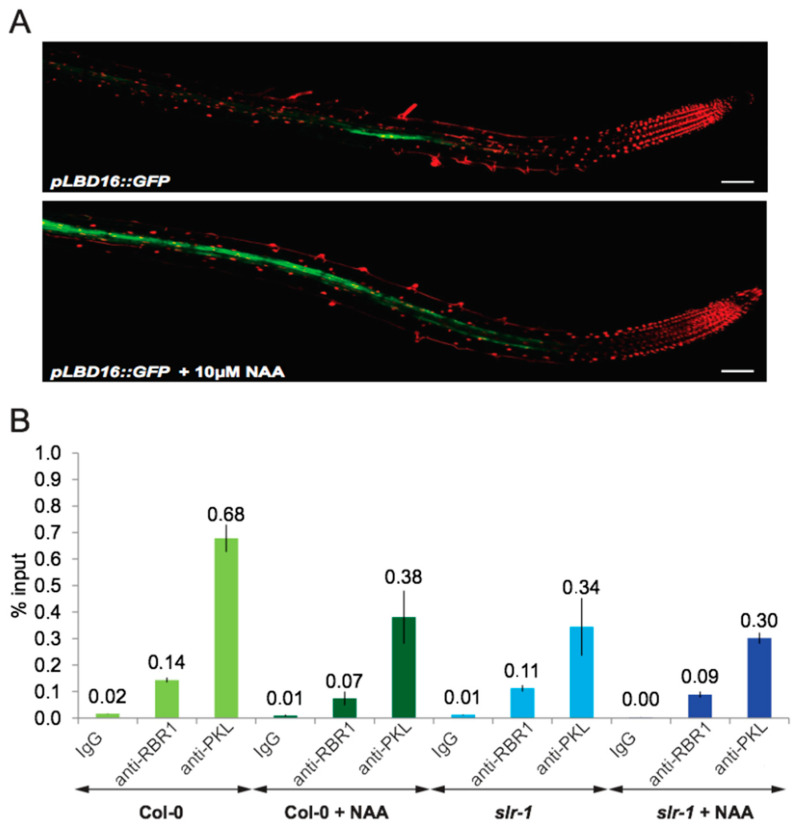
*LBD16* promoter activation upon NAA treatment. (**A**) Confocal microscopic images of 10 DAG Col-0 roots expressing the *pLBD16::GFP* construct with or without 10 μM NAA. Scale bar = 100 μm. (**B**) ChIP-qRT-PCR analysis of changes of RBR1 and PKL binding to *LBD16* promoter fragment (F4) in 10 DAG Col-0 and *slr-1* roots upon 10 μM NAA exposure. Bar graphs show quantification of qRT-PCR products from ChIP experiment using anti-RBR1 and anti-PKL antibodies, expressed as % input. Error bars represent means ± SD of the mean of three technical replicates.

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
