# Peer review of "Pickle Recruits Retinoblastoma Related 1 to Control Lateral Root Formation in Arabidopsis"

_ijms, 2021, doi:10.3390/ijms22083862_

Round 1

Reviewer 1 Report

Comments to the author:

This manuscript entitled “PICKLE recruits RETINOBLASTOMA RELATED 1 to Control Lateral Root Formation in Arabidopsis” by Krisztina Ötvös et al shows that the chromatin remodeling protein PICKLE negatively regulates auxin-mediated LR formation by interacting with RETINOBLASTOMA-RELATED 1 (RBR1) to repress the LATERAL ORGAN BOUNDARIES-DOMAIN 16 (LBD16) promoter activity. Their results focus on the new relationship between chromatin level regulation and auxin signaling to control formative cell divisions and lateral organ formation. And authors concluded that Inhibition of LR formation by PKL-RBR1 was counteracted by auxin indicating that in addition to auxin-mediated transcriptional responses, the fine-tuned process of LR formation was also controlled at the chromatin level in an auxin-signaling dependent manner. However, the provided data are not convincing to support this conclusion. Most of the figures were not properly arranged or assembled with poor quality. Major comments:

  1. Figure 1, 1A was not mentioned in the legend. 1C did not have the proper control.
  2. In Figure 2 they showed that “The RBR1 protein is present in xylem pole pericycle cells”, obviously its subcellular localization is not in the nucleus but in Figure 1 B the subcellular localization of the RBR1/PKL complex was in nucleus, how to explain this contradictory results?
  1. In figure 3B, did it represent the emerged lateral root density? As they show that “the PKL protein probably acts to repress either polar nuclear movement or the 244 subsequent asymmetric cell divisions”, the densities of lateral root primordium should be calculated at various stages, especially the early ones.
  2. In figure 3E, F, the “Phenotype of 10 DAG seedlings” and the “primary root length” were weird. I couldn’t believe that the seedlings grow.
  3. In figure 5B they should give a more detailed time-course analysis to show the expression of pLBD16::GFP during LRI.
  4. Since LBD16 function is required for the formative division of LR founder cells, if PKL-RBR model regulates LR through LBD16, authors need to examine the detailed LR development, not simply counting LR numbers.
  5. Since RBR controls cell division, how did author preclude the possibility that PKL-RBR regulates LR development through the regulation of cell division?

Author Response

ijms-1140682

Point-by-point response to Reviewer 01

This manuscript entitled “PICKLE recruits RETINOBLASTOMA RELATED 1 to Control Lateral Root Formation in Arabidopsis” by Krisztina Ötvös et al shows that the chromatin remodeling protein PICKLE negatively regulates auxin-mediated LR formation by interacting with RETINOBLASTOMA-RELATED 1 (RBR1) to repress the LATERAL ORGAN BOUNDARIES-DOMAIN 16 (LBD16) promoter activity. Their results focus on the new relationship between chromatin level regulation and auxin signaling to control formative cell divisions and lateral organ formation. And authors concluded that Inhibition of LR formation by PKL-RBR1 was counteracted by auxin indicating that in addition to auxin-mediated transcriptional responses, the fine-tuned process of LR formation was also controlled at the chromatin level in an auxin-signaling dependent manner. However, the provided data are not convincing to support this conclusion. Most of the figures were not properly arranged or assembled with poor quality. Major comments:

We would like to thank the reviewer for her/his opinion on our manuscript. Please note that the line numbers that we mention in this reply refer to our revised manuscript and not to the original manuscript. The Supplemental Figures and Information was also added in the revised version.

  1. Figure 1, 1A was not mentioned in the legend. 1C did not have the proper control.

Regarding 1A, thank you for drawing attention to the missing label (A) at the beginning of the legend. It has been corrected. In the original text file (A) was present in the legend. It must have been an editing problem. We are sorry for the mistake.

Regarding 1C, using pre-immune serum as negative control and input sample as positive control in an immunoprecipitation experiment is a widely accepted and highly recommended procedure. See for example the following citation by Hewitt et al (Hewitt et al., 2014) ‘In the instance of a non-commercial polyclonal antibody, pre-immune sera should be used…’.

  1. In Figure 2 they showed that “The RBR1 protein is present in xylem pole pericycle cells”, obviously its subcellular localization is not in the nucleus but in Figure 1 B the subcellular localization of the RBR1/PKL complex was in nucleus, how to explain this contradictory results?

We agree with the reviewer. The red fluorescent signal deriving from the pRBR1-RFP expressing plants (RFP signal in the nucleus in pericycle cells) could be mistaken by the red fluorescent signal deriving from Propidium Iodide (red fluorescence signal in the cell wall) staining and this could lead misinterpretation. To avoid any confusion regarding the localisation pattern of RBR1 Figure 2 has been replaced. We believe this replacement makes our statement – RBR1 protein is expressed in lateral root founder cells in the pericycle– stronger and visually more appealing.

  1. In figure 3B, did it represent the emerged lateral root density? As they show that “the PKL protein probably acts to repress either polar nuclear movement or the 244 subsequent asymmetric cell divisions”, the densities of lateral root primordium should be calculated at various stages, especially the early ones.

In Figure 3B and 3G ‘Lateral root density’ is shown; it was calculated by dividing the number of emerged lateral roots on the primary root with the length of the root.

Regarding counting the stages of lateral root development. The involvement of LBD16 in lateral root development is well described in the literature. As Lee et al (Lee et al., 2009) demonstrated the numbers of primordia of the lbd16 mutant were similar to those observed in the wild type but the numbers of emerged lateral roots of lbd16 single mutants were reduced significantly. That is the reason we used lateral root density in our lateral root phenotyping experiments.

  1. In figure 3E, F, the “Phenotype of 10 DAG seedlings” and the “primary root length” were weird. I couldn’t believe that the seedlings grow.

With all due respect, the seedlings did grow. We would have reported otherwise. Please have a look on Supplementary Figure S7 as well.

  1. In figure 5B they should give a more detailed time-course analysis to show the expression of pLBD16::GFP during LRI.

The idea behind this experiment was to quantify pLBD16::GFP expression at a certain/exact time-point after gravitropic bending to be able to investigate the involvement of RBR1 and PKL proteins in lateral root initiation.

  1. Since LBD16 function is required for the formative division of LR founder cells, if PKL-RBR model regulates LR through LBD16, authors need to examine the detailed LR development, not simply counting LR numbers.

We believe the answer given to reviewer’s comment number 3 explains why we haven’t done a detailed developmental analysis.

  1. Since RBR controls cell division, how did author preclude the possibility that PKL-RBR regulates LR development through the regulation of cell division?

We agree with the reviewer. We did not rule out this possibility.

As we mentioned in the introduction (lines 57-59), overexpression of CYCLIN D3;1, a known activating subunit of the G1/S regulator CDKA;1 kinase – that phosphorylates RBR1 activating the E2FA/B pathway - triggers only a few rounds of divisions in the pericycle in the slr-1 root but fails to initiate lateral root formation (Vanneste et al., 2005). So most likely the canonical cell cycle function of RBR1 does not play a crucial role in lateral root development.

On the other hand, since the function of the LBD16 protein is to control the first asymmetric division of the lateral root founder cells, one could easily interpretate our result that is RBR1-PKL complex directly regulates LBD16 promoter activity as RBR1 regulates division of the lateral root founder cells.

Hewitt, S.M., Baskin, D.G., Frevert, C.W., Stahl, W.L., and Rosa-Molinar, E. (2014). Controls for Immunohistochemistry: The Histochemical Society’s Standards of Practice for Validation of Immunohistochemical Assays. J. Histochem. Cytochem. 62, 693–697.

Lee, H.W., Kim, N.Y., Lee, D.J., and Kim, J. (2009). LBD18/ASL20 Regulates Lateral Root Formation in Combination with LBD16/ASL18 Downstream of ARF7 and ARF19 in Arabidopsis. Plant Physiol. 151, 1377–1389.

Vanneste, S., De Rybel, B., Beemster, G.T.S., Ljung, K., De Smet, I., Van Isterdael, G., Naudts, M., Iida, R., Gruissem, W., Tasaka, M., et al. (2005). Cell Cycle Progression in the Pericycle Is Not Sufficient for SOLITARY ROOT/IAA14-Mediated Lateral Root Initiation in Arabidopsis thaliana. Plant Cell Online 17, 3035–3050.

Reviewer 2 Report

The manuscript by Otvos and co-authors describes how PKL-RBR interaction impacts on the LR regulator LBD16. They build the story very carefully and clearly. All results are critically interpreted and discussed. I cannot add much to improve the manuscript.

Minor comments:

  • line 27: add a comma: "... that, in addition to... , the fine-tuned..."
  • in the introduction, the authors describe LR originating form XPP cells. However, this is a generalization. In grasses, LRs originate from phloem pole pericycle cells, and in leptosporangiate ferns, they originate from xylem pole endodermis cells. I think it is important to remind the readers of this variety
  • Fig2 is a bit too large, and would benefit from cropping the more relevant region, and add arrowhead at position of the RBR-RFP nuclei.
  • I am not too familiar with the transient plasmid ChIP, but one could think that the lack of enrichment could indicate that the experiment failed. Would there be positive controls for this otherwise elegant experiment?
  • line 387: for me "outside chromatin context" would also include a different position in the chromosome, eg. when making a stable transgene. Here, the lack of histones in plasmids is taken as "chromatin context".
  • line 575: I consider 2013 no longer as "recent"

Author Response

ijms-1140682

Point-by-point response to Reviewer 02

The manuscript by Otvos and co-authors describes how PKL-RBR interaction impacts on the LR regulator LBD16. They build the story very carefully and clearly. All results are critically interpreted and discussed. I cannot add much to improve the manuscript.

We would like to thank the reviewer for her/his opinion on our manuscript. We are very grateful for the reviewer’s comments and addressed them all in our response below. Please note that the line numbers that we mention in this reply refer to our revised manuscript and not to the original paper. The Supplemental Figures and Information was also added in the revised version.

Minor comments:

  • line 27: add a comma: "... that, in addition to... , the fine-tuned..."

Thanks for noticing this. It has been corrected. (line 23)

  • in the introduction, the authors describe LR originating form XPP cells. However, this is a generalization. In grasses, LRs originate from phloem pole pericycle cells, and in leptosporangiate ferns, they originate from xylem pole endodermis cells. I think it is important to remind the readers of this variety

Thank you for pointing out this mistake in our interpretation. We agree with reviewer’s comments on the different cellular origins of LRs. We followed the suggestion and corrected the introduction accordingly (line 26-37).

  • Fig2 is a bit too large, and would benefit from cropping the more relevant region, and add arrowhead at position of the RBR-RFP nuclei.

Figure 2 has been replaced. We believe data presented on the new version supports our statement – that the RBR1 protein is expressed in lateral root founder cells – better and visually more appealing.

  • I am not too familiar with the transient plasmid ChIP, but one could think that the lack of enrichment could indicate that the experiment failed. Would there be positive controls for this otherwise elegant experiment?

The primer pairs used in the plasmid ChIP experiments were carefully designed and tested beforehand. The primers were designed to be either genome or plasmid specific. With the combination of the primer sets we were able to distinguish between genome and plasmid specific signals. The ChIP results are presented as % input. The positive control is the input sample. All data is normalised to the input. If the experiment had not worked, we would not have been able to detect any signal in the input.

  • line 387: for me "outside chromatin context" would also include a different position in the chromosome, eg. when making a stable transgene. Here, the lack of histones in plasmids is taken as "chromatin context".

The text has been corrected (line 321-322).

  • line 575: I consider 2013 no longer as "recent"

It has been corrected (line 412).

Reviewer 3 Report

This manuscript reports very interesting and novel findings that PICKLE (PKL) recruits RBR1 to LBD16 promoter and represses the expression and lateral root formation, and that this repression is alleviated by auxin. Here, I raised some minor points to be addressed or corrected before publication as below.

Fig. 1C. It is better to add an WB result for RBR1 to confirm success of the immunoprecipitation assay.

Line 206. What is “XPP cells”? “lateral root founder cells” may be correct?

Some figures (Fig. 1D, Fig. 4BDC, Fig. 5C, Fig. 6 and Fig. 7B) lacked statistics. You should add any statistics to show significant differences.

Fig. 1. It is better to rephrase “amiGO RBR1” with amiRBR1.

Line 24. “PICKLE (PKL)” is right.

Lines 148-149. Delete “RETINOBLASTOMA-RELATED 1”.

Author Response

ijms-1140682

Point-by-point response to Reviewer 03

This manuscript reports very interesting and novel findings that PICKLE (PKL) recruits RBR1 to LBD16 promoter and represses the expression and lateral root formation, and that this repression is alleviated by auxin. Here, I raised some minor points to be addressed or corrected before publication as below.

We would like to thank the reviewer for her/his opinion on our manuscript. We took her/his comments are seriously and addressed them all.

Fig. 1C. It is better to add an WB result for RBR1 to confirm success of the immunoprecipitation assay.

Thank you for this comment of the reviewer. We agree that re-probing of the protein gel blot shown on Fig.1C. with the anti-RBR1 Ab would have been useful to confirm success of the IP reactions. Since both, the anti-RBR1 and anti-PKL polyclonal antibodies were raised in rabbits, re-probing required stripping of the gel blot after the first round of immunodetection. We have tried different stripping protocols (Tris-buffered SDS-DTT, 100mM glycine at pH 2.6 or 100mM citric acid) but our anti-RBR1 antibody was underperforming on stripped filters (developed a high and uniform background regardless of the stripping protocol). However, we have carefully tested the anti-RBR1 antibody in pilot IP reactions using our standard immunoprecipitation protocol to find an efficient and reproducible depletion of the RBR1 protein from diverse plant tissue extracts each time. Result of such a pilot experiment is illustrated below:

(please find the blot in the "Response to Reviewer 03.pdf" file)

In this pilot experiment 100µg of Arabidopsis root proteins were immunoprecipitated either with normal rabbit IgG (NR IgG), with pre-immune IgG (from the same animal that was used to raise the anti-RBR1 antibody) or with immune IgG (the purified anti-RBR1 antibody). Immunoprecipitated proteins as well as ¼ of the input (25µg) and supernatant fractions of the IP reactions (25µg proteins of each) were resolved and immunoblotted with the anti-RBR1 antibody.

We are confident that the presented immunoprecipitation result from root tissues together with the results obtained by two other methods (as presented in Fig. 1A and B) are clearly and without any doubt demonstrating the in vivo interaction of PKL with RBR1.

Line 206. What is “XPP cells”? “lateral root founder cells” may be correct?

Thanks for noticing this. XPP, means xylem pole pericyle. It has been added/corrected (line 206).

Some figures (Fig. 1D, Fig. 4BDC, Fig. 5C, Fig. 6 and Fig. 7B) lacked statistics. You should add any statistics to show significant differences.

Thanks for bringing this issue up. We corrected the following figures, added the requested significance and corrected the text accordingly: in Fig. 3. A, B and G (line 281), in Fig 4. B and C (lines 332-333 and 338-339), in Fig. 5C (lines 412-413), Fig. 6. A, B and D (lines 463-464, 470, 480-481). There is no panel “D” in Figure 1.

Fig. 1. It is better to rephrase “amiGO RBR1” with amiRBR1.

In Fig. 1 there is no “amigo RBR1”. Most probably the reviewer meant Fig. 3. It has been corrected (Fig. 3. C, D, F and G).

Line 24. “PICKLE (PKL)” is right.

Thanks for noticing this. It has been corrected (line 68 and 70).

Lines 148-149. Delete “RETINOBLASTOMA-RELATED 1”.

Thanks for noticing this. It has been corrected (line 148-149).
